# Vitamin D and Vitamin D Receptor Polymorphisms Relationship to Risk Level of Dental Caries

Marios Peponis [1], Maria Antoniadou [1,*], Eftychia Pappa [1], Christos Rahiotis [1] and Theodoros Varzakas [2,*]

1   Department of Dentistry, School of Health Sciences, National and Kapodistrian University of Athens, 11527 Athens, Greece; mariospep@gmail.com (M.P.); effiepappa@dent.uoa.gr (E.P.); craxioti@dent.uoa.gr (C.R.)
2   Department of Food Science and Technology, University of the Peloponnese, 24100 Kalamata, Greece
*   Correspondence: mantonia@dent.uoa.gr (M.A.); t.varzakas@uop.gr (T.V.)

**Featured Application: Factors influencing Vitamin D and its receptor polymorphisms on risk level of dental caries.**

**Abstract:** Dental caries is a multifactorial disease with multiple risk factors. Vitamin D levels (VDLs) and vitamin D receptor polymorphisms (VDRPs) have been investigated for this reason. The aim of this narrative review is to investigate the relation and the factors affecting vitamin D deficiency (VDD), VDRP, Early Childhood Caries (ECC) and Severe Early Childhood Caries (S-ECC) in children (primary and mixed dentition) and dental caries risk in adults (permanent dentition). Additionally, we present a model incorporating factors and interactions that address this relationship. Methods: Three databases (PubMed/MEDLINE, Web of Science, Cochrane Library) were comprehensively searched until 17 January 2023 using the following keywords: "vitamin D", "vitamin D receptor polymorphism", "dental caries", and "dental caries risk", finding 341 articles. Two reviewers searched, screened, and extracted information from the selected articles. All pooled analyses were based on random-effects models. Eligibility criteria were articles using dmft/DMFT diagnostic criteria with calibrated examiners, probability sampling, and sample sizes. We excluded studies conducted on institutionalized patients. A total of 32 studies were finally used. Results: In most studies, *TaqI*, *FokI*, and *BsmI* polymorphisms affected the prevalence of dental caries. A strong correlation between ECC, S-ECC, and the prevalence of dental caries was reported in association with VDD and maternal intake of VD in primary dentition. Regarding the influence in mixed dentition, the results were found to be inconclusive. A slight positive influence was reported for permanent dentition. Conclusions: Factors affecting caries risk were maternal intake, socioeconomic factors, and level of VD. There is a certain need for more well-conducted studies that will investigate the association between VDR gene polymorphisms and the prevalence of dental caries in mixed and permanent dentition, specifically in adult patients.

**Keywords:** vitamin D receptor; polymorphism; dental caries; dentistry; prevalence; prevention; dmft/DMFT index

## 1. Introduction

Dental caries is a multifactorial disease with multiple risk factors [1]. Vitamin D levels (VDLs) and vitamin D receptor polymorphisms (VDRPs) have been investigated for this reason [2]. Vitamin D is a fat-soluble steroid hormone which regulates calcium and phosphorus levels through the intestine [3]. This vitamin can be found in the human body as vitamin D3 (cholecalciferol) and vitamin D2 [2]. Both are converted to 25-hydroxyvitamin D (25(OH)D), which acts as a biological marker for vitamin D levels in serum. When 25(OH)D reaches the kidneys, it is converted to calcitriol (1,25(OH)2D), the most active form of vitamin D, with a shorter half-life [3].

Vitamin D deficiency (VDD) is defined when levels of 25(OH)D are below 20 ng/mL (50 nmol/L) and there is an insufficiency between 21–29 ng/mL (50–75 nmol/L) [4]. VDD has, therefore, abnormality in calcium, phosphorus, and bone metabolism [4] and is associated with increased risk of neoplastic, metabolic, and immune disorders [5,6]. Tooth mineralization co-occurs to skeletal mineralization, so disturbances in mineral metabolism affect bone and teeth. As VD has a crucial role in tooth and bone mineralization, low levels of VD may lead to a defective and hypomineralized tooth [6]. The main mechanism related to severe VDD causes hypophosphatemia and hypocalcemia with secondary hyperparathyroidism [7]. Hyperparathyroidism, in turn, stimulates renal production of 1,25(OH)2D and absorption of calcium in intestines. It increases bone turnover and may lead to increased serum levels of calcium ions and decreased serum levels of inorganic phosphate [8]. Therefore, proper mineralization of teeth is inhibited due to the loss of VD signaling pathways in tooth cells as concentrations of calcium ions and phosphate ions are low [9–11]. The biological activity of VD is modulated and modified by the vitamin D receptor (VDR) protein. VDR is responsible for the expression of many genes involved in cellular proliferation and differentiation, calcium–phosphate homeostasis, and immune response [12]. VDR is regulated by the VDR gene, whose polymorphisms affect the function of the VDR protein. Polymorphisms are genetic variations in non-coding parts of the gene (introns) or in the exonic parts of the DNA that influence at least 1% of the population [13]. Changes in introns do not influence the protein product but can modify the degree of gene expression. On the other hand, changes in exonic parts of the gene affect the protein sequence, except for synonymous polymorphisms, which are alterations in exonic parts that do not affect the protein structure [13]. These variations are responsible for the creation or removal of restriction enzyme sites in DNA, which, in turn, create DNA fragments with various lengths [14]. The most common polymorphisms (VDRP) of the VDR gene, which have been related to oral and systemic conditions, are *BsmI (rs1544410), Taql (rs731236), BglI (rs739837), ApaI (rs7975232), FokI (rs10735810)*, and *FokI (rs2228570)* [14,15].

VDD has been associated with Early Childhood Caries (ECC), and Severe Early Childhood Caries (S-ECC) [10]. ECC is a global health problem, affecting almost half of preschool children, defined as the presence of one or more decayed, missing, or filled primary teeth in children aged 71 months (5 years) or younger [11]. ECC has a multi-factorial etiology, including susceptible teeth due to colonization with high levels of cariogenic bacteria, such as *S. mutans*, enamel hypoplasia, and sugar metabolism, and by bacteria that produce acid, which, in turn, demineralizes tooth structure [10]. A subtype of ECC is Severe Early Childhood Caries (S-ECC). S-ECC is defined as any sign of smooth-surface caries in a child under the age of three and, from ages three to five, one or more cavitated, missing, or filled smooth surfaces in primary maxillary anterior teeth or a decayed, missing, or filled score greater than or equal to 4 (for 3 years of age), 5 (for 4 years of age), or 6 (for 5 years of age) [11].

As awareness of VDD has increased among patients and the health community, many authors have conducted clinical trials to study a potential association between VDD and dental caries [12–14]. In addition, there is a systematic review investigating vitamin D receptor (VDR) gene polymorphisms in relation to increased risk of dental caries only in children [15–17]. The aim of this study was to further investigate the relation between VDD, VDRP, ECC, and S-ECC in children (primary and mixed dentition) and dental caries risk in adults (permanent dentition). Additionally, we present a model incorporating factors and interactions that address this relationship.

## 2. Materials and Methods

For this study, an extended review of the relevant literature was performed. The algorithm of the search was "vitamin D and Polymorpisms and dental caries and abstract and vitamin D receptor". Records have been selected from three databases (PubMed/Medline, Web of Science, Cochrane Library). Two reviewers searched, screened, and extracted information from the selected articles. All pooled analyses were based on random-effects models. Eligibility criteria were articles using dmft/DMFT diagnostic criteria with cali-

brated examiners, probability sampling, and sample sizes. We excluded studies conducted on institutionalized patients. A total of 341 articles were reported in the initial search, of which 104 were removed as duplicates. A total of 40 articles were excluded for other reasons, such as inadequate methodology or irrelevance to our search content through abstract evaluation. In the second phase of the study, from the remaining 197 articles, 145 were excluded due to inadequate methodology. In the third phase of the screening, 32 studies were finally included in the review, as seen in detail in the following Prisma flow chart (Figure 1).

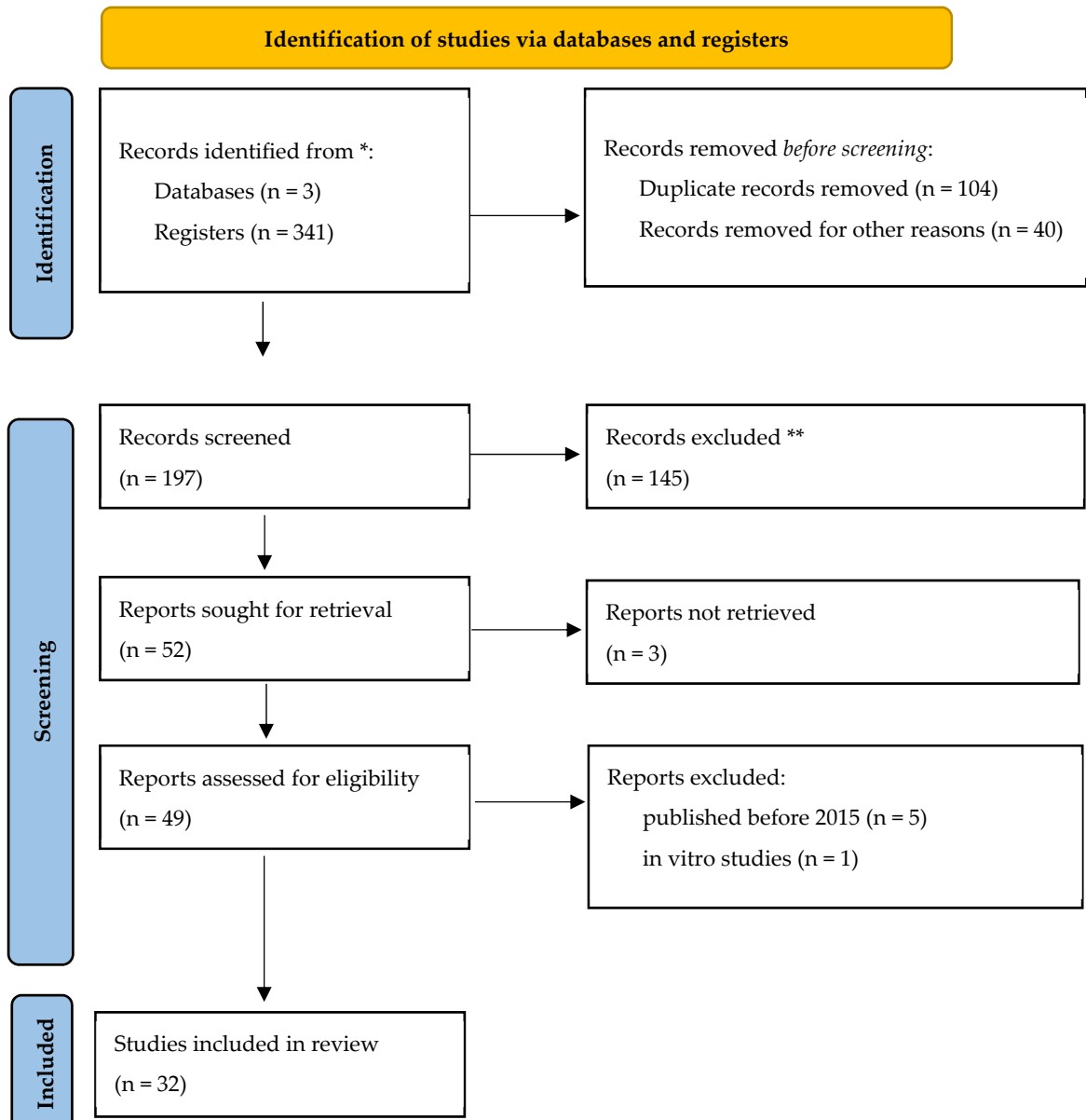

**Figure 1.** Prisma flow chart of the present study. * Eligibility criteria were articles using DMFT diagnostic criteria with calibrated examiners, probability sampling, and sample sizes.** We excluded studies conducted on institutionalized patients.

### 3. Results

*Vitamin D Receptor Gene Polymorphisms*

To discuss possible relations between VDPP and caries, we needed to report on the possible alleles of the polymorphisms of the VDR gene, presented in Table 1.

**Table 1.** Different types of alleles and genotypes of the VDR gene polymorphisms.

| Genotype | *ApaI*<br>*(rs7975232)* | *TaqI*<br>*(rs731236)* | *FokI*<br>*(rs10735810)* | *FokI*<br>*(rs2228570)* | *BglI*<br>*(rs739837)* | *BsmI*<br>*(rs1544410)* |
|---|---|---|---|---|---|---|
| Homozygous dominant | AA | TT | FF | FF | BB | BB |
| Heterozygous | Aa | Tt | Ff | Ff | Bb | Bb |
| Homozygous recessive | Aa | tt | Ff | Ff | bb | bb |

From our findings we can support the fact that all relevant studies are differentiated in both methodology and sample size, thus revealing differences in the hypothesis that certain polymorphisms may play a specific role in caries risk assessment (Table 2).

For example, Kong et al. found a correlation between the *BsmI* polymorphism containing the Bb genotype and increased risk of caries in deciduous teeth of Chinese children [18]. On the other hand, two studies in the Chinese population did not show a statistically significant difference between the caries-free and caries-active groups [19,20]. In numerous studies, no statistically significant difference was found for a higher caries prevalence in the population with *ApaI* polymorphism [18–21]. In the case of the *TaqI* polymorphism, one study suggested the allele 't' as a possible genetic factor for determining dental caries in Chinese individuals because the 't' allele was more frequent in the caries-active group [22]. Similar results were found by Cogulu et al. [19] concerning the *TaqI* polymorphism, showing a statistically significant difference in *TaqI* genotypes between the case and control groups. On the other hand, many other case–control studies did not report significant differences in allele and genotype frequency of the *TaqI* polymorphism between individuals with and without caries [18–20,23,24]. Further, the *BgII* polymorphism has been studied by two research groups in the Brazilian population. The authors concluded that the *BgII* polymorphism was not associated with an increased risk of dental caries in the respective population [25,26]. From all polymorphisms studied, the *FokI* polymorphism, distinguished in *rs10735810* and *rs2228570*, is reported to play a significant role in the issue studied here. Although two studies supported that different alleles and genotypes of *rs10735810* were not associated with tooth decay in Chinese and Turkish populations [18,19], two other studies found a statistically significant difference when the *FokI rs10735810* genotype was present in Chinese and Brazilian populations [20,25]. In terms of the *rs2228570* polymorphism, no correlation was found regarding a higher dental caries risk in the Brazilian population [25,26]. A systematic review and meta-analysis compared the influence of the *ApaI*, *FokI (rs10735810)*, *TaqI*, *BsmI*, *FokI (rs2228570)*, and *BglI* polymorphisms of the VDR gene on dental caries risk [15]. Based on nine studies, the meta-analysis reported an association between the *FokI (rs10735810)* polymorphism and dental caries risk in children, with the f allele and the ff genotype reported as having a protective role. Additionally, authors found that there was no association between the *ApaI (rs7975232)*, *TaqI (rs731236)*, *BsmI (rs1544410)*, *FokI (rs2228570)*, and *BglI (rs739837)* polymorphisms of the VDR gene and the prevalence of dental caries in children [15]. The authors claimed that the protective role of the f allele and ff genotype might be due to their interactions with co-transcription factors and location [15]. It is then concluded that there is a lack of consistency among the case–control studies reported here, their results related to the different VDR gene polymorphisms, and their influence on dental caries risk [27–29]. This fact is attributed to the statistical heterogeneity among studies, the small number of existing relative studies, and the small sample sizes. Thus, there is a significant need for additional, well-conducted research that will investigate the association between VDR gene polymorphisms and the prevalence of dental caries in mixed and permanent dentition, specifically in adult patients.

**Table 2.** Studies investigating correlation between VDR polymorphisms and prevalence of dental caries.

| Authors Publication Year | *ApaI* rs7975232 | *TaqI* rs731236 | *FokI* rs10735810 | *FokI* rs2228570 | *Cdx2* rs11568820 | *BglI* rs739837 | *BsmI* rs1544410 | Type of Study/Country | Results |
|---|---|---|---|---|---|---|---|---|---|
| Hu et al., 2015 [22] | | + | | | | | | Case (264)–control (219) study/China Permanent dentition | • 't' allele more frequent in case group (7%) than control (2.1%) (*p* = 0.0003) <br> • allele 't' may be a genetic factor for the determination of dental caries individuals in Chinese population |
| Holla et al., 2017 [23] | | + | | | | | | Case (235)–control (153) study/Czech Permanent dentition | No significant differences in allele and genotype frequency of *TaqI* between case and control group |
| Cogulu et al., 2016 [19] | + | + | + | | + | | | Case (57)–control (38) study/Turkey Primary dentition | • statistically significant difference between *TaqI* genotypes between control and case group (*p* = 0.029) <br> • no statistically significant difference for genotypes of *ApaI*, *FokI*, *Cdx2* |
| Kong et al., 2017 [18] | + | + | + | | | | + | Case (249)–control (131) study/China Primary dentition | • increased risk of deciduous tooth decay in Bb genotype of *BsmI* <br> • *ApaI, TaqI, FokI* polymorphisms are not associated with deciduous tooth decay |
| Yu et al., 2017 [20] | + | + | + | | | | + | Case (200)–control (200) study/China Permanent dentition | • C allele frequency of the FokI VDR polymorphism was significantly increased in the case group (*p* < 0.001) <br> • *BsmI, TaqI, ApaI* showed no statistically significant difference between case and control group <br> • *FokI* gene polymorphism might be related with susceptibility to permanent decayed teeth in Chinese adolescent |

**Table 2.** *Cont.*

| Authors Publication Year | *ApaI* rs7975232 | *TaqI* rs731236 | *FokI* rs10735810 | *FokI* rs2228570 | *Cdx2* rs11568820 | *BglI* rs739837 | *BsmI* rs1544410 | Type of Study/Country | Results |
|---|---|---|---|---|---|---|---|---|---|
| Qin et al., 2019 [21] | + | + | + | | + | | + | Case (304)–control study (245)/China Primary dentition | • *FokI* genotype was statistically greater in the high caries risk group than moderate risk and control groups (*p* = 0.028)<br>• The use of VDR polymorphisms as markers for increased risk of dental caries in Chinese children is not reliable |
| Barbosa et al., 2020 [25] | | | | + | | + | | Case–control study/Brazil Permanent dentition | *FokI, BgII* VDR gene polymorphisms were not associated with dental caries |
| Aribam et al., 2020 [24] | | + | | | | | | Case (60)–control (60) study/India Permanent dentition | • No statistically significant differences between 'TT', 'Tt', 'tt' genotypes among case and control group<br>• Individuals with 't' allele and 'tt', 'Tt' genotypes might be susceptible to dental caries |
| Fatturi et al., 2020 [26] | | | | + | | + | | Case–control study/Brazil Permanent dentition | • No relation between MIH, HPSM, and dental caries with *BbII, FokI* VDR gene polymorphisms<br>• Higher prevalence of MIH was observed when an individual carried at least one 'G' allele (*p* = 0.03) |

**Table 2.** *Cont.*

| Authors Publication Year | *ApaI* *rs7975232* | *TaqI* *rs731236* | *FokI* *rs10735810* | *FokI* *rs2228570* | *Cdx2* *rs11568820* | *BglI* *rs739837* | *BsmI* *rs1544410* | Type of Study/Country | Results |
|---|---|---|---|---|---|---|---|---|---|
| Sadeghi et al., 2021 [15] | + | + | + | + | | + | + | Systematic review, meta-analysis Primary and permanent dentition | • Out of nine studies included in the meta-analysis, there was no association between *TaqI, Apal, Bsml, FokI (rs2228570), BglI* VDR polymorphisms and dental caries risk<br>• Protective role of f allele and ff genotype *Fokl (rs10735810)* VDR polymorphism in relation to dental caries |
| Nireeksha et al., 2022 [7] | | | | + | | | | Case (239)–control (138) study/India Permanent dentition | • Salivary vitamin D levels were higher in the control group (caries-free) ($p < 0.001$)<br>• T allele of Fokl VDR polymorphism significantly associated with having active caries, while C allele with being caries-free<br>• 2.814-fold increased possibility of TC genotype of *rs2228570* to be caries-active and 3.116-fold increased possibility of TT genotype to be caries-active |

The effect of vitamin D serum levels on the prevalence of dental caries has also been studied in many scientific papers among different ages, populations, and dentitions. Table 3 presents studies correlating vitamin D serum levels with the prevalence of dental caries.

From the studies presented in Table 3, we can report on 10 studies addressing the issue on primary dentition, 8 on mixed, 3 on permanent, and one on mixed and permanent dentition. Results from the above-mentioned 10 studies on primary dentition indicate an association between prenatal levels of 25(OH)D and caries risk in primary dentition. It is mentioned that maternal 25(OH)D, during pregnancy, diffuses across the placental barrier, while 25(OH)D cord blood concentrations are 75–90% of maternal concentrations at delivery [30]. Thus, in four out of ten studies, there was an inverse relation between maternal intake of VD and dental caries in children. In favor of this correlation, we refer to the study of Tanaka et al., which examined 1210 Japanese mother–child pairs and agreed on the issue [31]. This inverse relation was also verified by another study, in which the authors found that insufficient levels of 25(OH)D (<50 nmol/L) during the third semester of pregnancy is associated with higher caries experience in primary teeth by the age of six [32]. In a third study, data were extracted from an extensive cohort survey in Austria on VDLs during the 12th week of gestation and after gestation at 4 and 8 years of age, while a dental examination was held at 6 and 10 years of age. Although sugar consumption and incorrect brushing were the main factors associated with dental caries in children participating in the study, low levels of 25(OH)D (<20 ng/mL) during pregnancy magnified the effect of cariogenic parameters. For this reason, the authors claimed that supplementing VD for pregnant women and children could be an option [33]. These findings agree with a 6-year follow-up randomized clinical trial, which concluded that supplementation with a high dose of VD during pregnancy leads to a lower probability of enamel defects in the offspring [34]. On the contrary, Schroth et al. did not find a statistically significant difference when they administered prenatally two oral doses of 50,000 (IU) of VD in the prevalence of Early Childhood Caries [35].

Furthermore, nine studies have investigated the relationship between VD deficiency and ECC or S-ECC. More specifically, there was a negative correlation between VDD and increased risk of ECC [36–45]; for example, Singleton et al. [41] stated that deficient cord blood 25(OH)D levels (<30 nmol/L) had a significant impact on the dmft in the follow-up period between 12 and 35 months of infants' age, with dmft scores that were double in relation to nondeficient concentrations. In the same study, no significant association between insufficient levels of 25(OH)D (<50 nmol/L) and ECC was reported. In another study, with a significant sample size of 1510 Chinese children, VD deficiency and insufficiency led to an increased risk of ECC [38]. Chhonkar et al. suggested supplementation of VD to prevent dental caries because VDD is a significant risk factor for dental caries in children [39]. In addition to VDD, children with S-ECC showed significantly lower levels of Ca and serum albumin and higher levels of PTH [40]. Deficient and insufficient levels of VD led to a higher odds ratio for S-ECC in children compared to optimal VD levels, according to two studies that have been conducted in Canada and North America [43,44]. Only one case–control study reported levels of VD in children without caries comparatively higher than in children with severe caries at a mean age of 40.82 months, even if the mean levels of VD were close to optimal [42]. A cross-sectional study with a sample of 1638 children from Poland concluded that the prevalence of ECC and S-ECC was significantly lower in children receiving VD supplementation. The authors also reported that children and mothers of higher education received VD supplementation and had fewer caries, highlighting the socioeconomic background of ECC [45]. In six out of nine studies addressing the issue, it is reported that socioeconomic factors, and especially the mothers' educational background, have a negative impact on caries risk and a positive effect on levels of VD.

Furthermore, many studies have investigated the relation between 25(OH)D levels and dental caries risk in mixed dentition, of which eight are included in this study. In all studies included here, there seems to be a weak indication that improving children's VD status might be an additional preventive measure against caries during the period of

mixed dentition. In more detail, a cross-sectional study with a sample of 1017 Canadian children from 6 to 11 years of age stated that optimal levels of 25(OH)D (>75 nmol/L) had a significantly lower odds ratio for caries [46]. Further, Navarro et al. reported a weak association between the risk of dental caries and 25(OH)D serum, and the authors did not recommend supplementation of VD as a preventive measure for managing dental caries [47]. A weak inverse correlation between VD and dental caries risk in mixed dentition was found also in another study with a sample of 206 Swedish children [48]. Supplementation of VD and fluoride during the entire first year of a child's life led to a lower probability of caries-related restorations at the age of 10 in primary teeth in relation to children who received the same supplementation for less than 6 months [49]. A cohort study with a sample of 335 Portuguese children and a mean age of 7 years reported that 25(OH)D levels less than 30 ng/mL were associated with advanced dental caries in permanent teeth, while no statistically significant difference was found in terms of 25(OH)D levels and dental caries risk in mixed dentition [50]. The same results, regarding mixed dentition, were reported in a study with a sample of 121 Polish children with growth hormone deficiency. There was no statistically significant impact of VD on the mean DMFT index, but the authors recommended VD as a potentially effective agent in reducing dental caries, especially for growth hormone deficiency patients [51].

Apart from primary and mixed dentitions, a possible correlation between levels of 25(OH)D and dental caries risk has been investigated for permanent dentition. In our study, we report on three studies regarding permanent dentition and one with both mixed and permanent ones. From our findings in all four relevant studies, it seems that permanent dentition presents a higher correlation with VD deficiency than the mixed one. To support this, we can report that in a cross-sectional study with a sample of 2579 American adolescents, VD deficiency in permanent dentition has a limited but statistically non-significant association with caries in permanent dentition of adolescents [52]. A cross-sectional study on 1688 children from Korea found that children with serum 25(OH)D levels lower than 50 nmol/L were 1.29 times more likely to develop dental caries and that 25(OH)D concentration and DMFT were negatively correlated. However, the authors reported that there is difficulty in confirming the association between dental caries experience and 25(OH)D levels [53]. Another study that has investigated permanent dentition and 25(OH)D levels was conducted in the USA with 4244 participants with a mean age of 51.22 years. This study stated that the severely deficient group was strongly correlated with dental caries and that 25(OH)D is significantly associated with the prevalence of dental caries among US adults [54].

Concerning the VD supplementation for caries prevention in permanent dentition, we found data in all three relevant studies. Firstly, a systematic review of control clinical trials studied the influence of VD supplementation in two different forms (vitamin D2, D3) on dental caries prevention [55]. The results showed that VD supplementation reduced the risk of dental caries by about 47%, but with low certainty [55]. In two different mendelian randomization studies, the results failed to find a strong and statistically significant causal relationship between vitamin D concentrations and risk of dental caries [56,57].

Driven by the analysis performed in this study, we present a relevant model of the VD, VDRP, and dental caries equation. The model was designed with the Vensim program (Vensim PLE 8.1.0), showing factors influencing the equation. Vensim, developed by Ventana Systems (a simulation system employing causal tracing; US Patent Application EP19910909851, 26 February 1991), is precision simulation software. It primarily shows continuous simulation (system dynamics), with agent-based modeling capabilities. This modeling language supports arrays (subscripts) and permits mapping among factors. The built-in allocation functions satisfy, in our case, constraints that may not be met by conventional approaches (Figure 2).

**Table 3.** Studies investigating the relation between 25(OH)D levels and dental caries.

| Authors Publication Year | Type of Study | Population | Age | Dentition | Results | Conclusions |
|---|---|---|---|---|---|---|
| Schroth et al., 2013 [40] | Case–control study | • 144 Canadian children with S-ECC <br> • 122 Canadian children caries-free | • 42 +/− 11.9 (case group) <br> • 39.4 +/− 16.3 (control group) | Primary | • More children receiving vitamin D drops were in caries-free group ($p < 0.001$) <br> • 25(OH)D levels were significantly lower in S-ECC group ($p < 0.001$) <br> • Significantly more children with S-ECC had 25(OH)D < 75 nmol/L compared to control group ($p = 0.006$) | • Significantly lower vitamin D in children with S-ECC compared to caries-free control group <br> • Children with S-ECC show significantly lower levels of Ca, serum albumin, and higher levels of PTH compared to caries-free control group |
| Tanaka et al., 2015 [31] | Prospective study | 1210 Japanese mother–child pairs | 36–46 months postnatal for the evaluation | Primary | • Linear relationship between vitamin D intake and log-odds of dental caries <br> • Inverse relationship between maternal intake of vitamin D and dental caries in children | Association between higher maternal vitamin D intake during pregnancy and a reduced risk of dental caries |
| Schroth et al., 2016 [46] | Cross-sectional study | 1017 Canadian children | 6 to 11 years | Mixed | • 25(OH)D levels > 75 nmol/L in children had significantly lower odds ratio for caries (OR = 0.57, 95% CI 0.39 to 0.82) <br> • Levels >50 nmol/L had lower odd ratio for caries (OR = 0.56, 95% CI 0.39 to 0.80) <br> • Levels of 25(OH)D > 50 nmol/L were significantly and independently related to lower adjusted odds for caries (OR = 0.46, 95% CI 0.26 to 0.83) <br> • Concentrations > 75 nmol/L were significantly and independently related to lower dmft/DMFT scores | Association between caries and lower serum vitamin D in a sample of Canadian children |

**Table 3.** *Cont.*

| Authors Publication Year | Type of Study | Population | Age | Dentition | Results | Conclusions |
|---|---|---|---|---|---|---|
| Kühnisch et al., 2017 [49] | Clinical trial | • 406 children | • Enrollment at the 1st year of age <br> • Dental examination at the age of 10 | Mixed | • Supplementation of vitamin D and fluoride during the entire first year of life led to significantly lower probability of having caries-related restorations in primary teeth in relation to those who received supplementation for less than 6 months <br> • No statistically significant difference between supplementation and prevalence of MIH | • Preventive effect of fluoride/vitamin D supplementation over the first year of life in the primary dentition |
| Chhonkar et al., 2018 [39] | Case–control study | • 60 Indian children | • 4.4 +/− 0.89 years (case group) <br> • 4.5 +/− 1.1 years (control group) | Primary | • Statistically significant difference in mean levels of 25(OH) vitamin D levels between two groups (*p* < 0.0001) <br> • Statistically significant inverse correlation between vitamin D levels and S-ECC (*p* < 0.0001) | • VDD is an important risk factor for incidence and severity of dental caries in children <br> • Prevention of dental caries in children by supplementing vitamin D and by preventing VDD |
| Deane et al., 2018 [44] | Case–control study | • 266 children in Canada | • Mean age 40.8 +/− 14.1 months | Primary | • Association between 25(OH)D levels below 75 nmol/L and S-ECC (*p* = 0.007) <br> • Combined deficiency of hemoglobin and 25(OH)D showed higher odds for dental caries (*p* < 0.001) <br> • Combined deficiency of iron or iron deficiency anemia with 25(OH)D levels < 75 nmol/L showed higher odds for dental caries (*p* < 0.001, *p* = 0.004, respectively) | • Due to the low frequency of children with S-ECC and combined deficiencies, authors do not suggest laboratory investigation <br> • Dietary history and dietary advice may have a helpful role in children with S-ECC, especially those of lower income |

**Table 3.** *Cont.*

| Authors Publication Year | Type of Study | Population | Age | Dentition | Results | Conclusions |
|---|---|---|---|---|---|---|
| Gyll et al., 2018 [48] | Intervention study | • 206 children from Sweden | • 6 years old at the baseline<br>• 8 years old for examination | Mixed | • Weak inverse association between vitamin D levels at 6 years of age and caries 2 years later (OR = 0.96, $p = 0.024$)<br>• Multivariate projection regression showed insufficient vitamin D concentration correlated with caries, while higher levels of 25(OH)D were correlated with being caries-free<br>• Vitamin D positively correlated with the levels of LL37 in saliva | • Vitamin D was not related to enamel defects on permanent incisors and molars<br>• Association between 25(OH)D and LL37 levels<br>• Negative correlation between vitamin D and caries; however, a small study group and weak association |
| Wójcik et al., 2019 [51] | | • 121 polish children and adolescents with growth hormone deficiency | • 6 to 18 years of age | Mixed | • Statistically significant impact of vitamin D levels on the average DMFT index in children and adolescents from rural areas ($p = 0.049$)<br>• No statistically significant impact of vitamin D levels on mean DMFT index ($p = 0.73$) | • Decrease in dental caries when vitamin D levels are increased in children who live in rural areas and are treated for growth hormone deficiency<br>• In urban areas, vitamin D supplementation and intensification of dental care is needed in children with growth hormone deficiency<br>• Promotion of vitamin D as a potentially effective agent in reducing dental caries in patients with growth hormone deficiency |
| Akinkugbe et al., 2018 [52] | Cross-sectional | • 2579 American adolescents | Two age groups:<br>• 12–14<br>• 15–19 | Permanent | Participants with insufficiency and deficiency of vitamin D had non-statistically significant adjusted estimates of 1.02 (0.72, 1.44) and 1.23 (0.7, 2.16), respectively, for caries experience | Deficiency of vitamin D appears to have limited but statistically non-significant association with adolescent caries |

**Table 3.** *Cont.*

| Authors Publication Year | Type of Study | Population | Age | Dentition | Results | Conclusions |
|---|---|---|---|---|---|---|
| Kim et al., 2018 [53] | Cross-sectional study | • 1688 Korean children | 10–12 years of age | Permanent | • 25(OH)D levels lower than 50 nmol/L had a higher incidence with respect to dental caries in the permanent dentition and permanent first molar ($p = 0.012$, $p = 0.006$, respectively)<br>• 25(OH)D concentration and DMFT were negatively correlated ($p < 0.01$) | • There is a difficulty in confirming the association between dental caries experience and 25(OH)D levels<br>• Insufficiency of 25(OH)D might be associated with dental caries |
| Singleton et al., 2019 [41] | Cohort study | • 76 prenatal Alaskan mother–infant pairs with prenatal blood were examined<br>• 57 Alaskan infants were examined by measuring cord blood | Two periods for follow-up of the infants:<br>• 12 to 35 months<br>• 36 to 59 months | Primary | • Significant difference in 12–35 months age group in DMFT score with deficient cord blood 25(OH)D ($p = 0.002$)<br>• Negative correlation between cord blood 25(OH)D levels and DMFT in the 12 to 35 months age group (R = −0.37, $p = 0.016$)<br>• No significant difference in mean DMFT for age groups 12–35, 36–59 months of age with prenatal 25(OH)D levels below or above 50 nmol/L | • Prenatal vitamin D levels may have an impact on the primary dentition and the risk of developing ECC<br>• Improving vitamin D levels in pregnant women may affect ECC rates in their infants |
| Schroth et al., 2020 [35] | Prospective cohort | • 283 mothers in Canada<br>• 175 for the follow-up | | Primary | • No significant difference between intervention group (2 oral prenatal doses of 50,000 (IU) vitamin D) and control group in prevalence of ECC ($p = 0.21$)<br>• Inverse correlation between dt scores and 25(OH)D ($p = 0.001$)<br>• Age, socioeconomic factor index, and enamel hypoplasia were significantly and independently associated with dt, while vitamin D supplementation was not | • Significant inverse relation between the number of teeth with caries and the levels of 25(OH)D<br>• Supplementation with vitamin D during pregnancy did not influence the prevalence of ECC |

**Table 3.** *Cont.*

| Authors Publication Year | Type of Study | Population | Age | Dentition | Results | Conclusions |
|---|---|---|---|---|---|---|
| Zhou et al., 2020 [54] | Cross-sectional study | • 4244 participants in the USA | • 20–80 years of age, mean age 51.22 +/− 17.86 | Permanent | • In three different models with different covariates (crude model, model 1, model 2) there was a negative correlation between 25(OH) serum levels and dental caries<br>• Severely deficient group was strongly associated with dental caries | • 25(OH)D concentration is significantly associated with prevalence of dental caries among US adults |
| Jha et al., 2021 [42] | Case–control study | 266 children from India | Mean age 40.82 +/− 14.09 months | Primary | Children with severe caries had significantly lower vitamin $D_3$ in very young childhood (68.87 +/− 28.04 vs. 82.89 +/− 31.12 nmol/L, $p < 0.001$) | Levels of vitamin $D_3$ in children without caries were comparatively higher than in children with severe caries |
| Navarro et al., 2021 [47] | Prospective cohort study | 5257 multi-ethnic children | Mean age 6.1 (4.8–9.1) | Mixed | • Severe deficiencies of 25(OH)D (<25 nmol/L) prenatally and in early childhood associated with higher prevalence of dental caries than children with sufficient concentrations (>75 nmol/L)<br>• Longitudinal association between low early childhood 25(OH)D serum concentrations and caries at 6 years of age<br>• Children with genetically predisposed low serum 25(OH)D concentrations do not show higher risk of developing caries in primary dentition | • Weak association between risk of dental caries and 25(OH)D serum concentrations<br>• Authors do not suggest vitamin D supplementation as a preventive measure for managing dental caries |

**Table 3.** *Cont.*

| Authors Publication Year | Type of Study | Population | Age | Dentition | Results | Conclusions |
|---|---|---|---|---|---|---|
| Silva et al., 2021 [50] | Cohort study | 335 Portuguese children | Mean age 7 years | Mixed and permanent | • Levels of 25(OH)D below 30 ng/mL lead to more dental caries in permanent teeth ($p = 0.016$, $p = 0.034$ for advanced dental caries) • No statistically significant difference in terms of 25(OH)D levels and dental caries risk in mixed dentition ($p = 0.288$) | • 25(OH)D levels < 30 ng/mL are associated with advanced dental caries in 7-year-old children's permanent teeth • In mixed dentition, social and behavioral factors influence the prevalence of dental caries in the examined Portuguese sample • Sufficiency of vitamin D might have an additionally protective role in terms of dental caries in permanent teeth |
| Williams et al., 2021 [43] | Case–control study | 144 caries-free children 200 children with S-ECC from North America | 42.1 +/− 14.6 months | Primary | • Children with S-ECC had significantly lower mean levels of 25(OH)D than the control group ($p < 0.001$) • Children with deficient and adequate concentrations of 25(OH)D were significantly more likely to have S-ECC than children with optimal levels (OR = 10.1, $p < 0.001$ and OR = 1.8, $p = 0.01$, respectively) | • Significant and independent association between caries and 25(OH)D levels • Children with optimal levels of 25(OH)D had lower odds for S-ECC |
| Olczak-Kowalczyk et al., 2021 [45] | Cross-sectional study | 1638 children from Poland | 3 years of age | Primary | • Significantly lower prevalence of ECC/S-ECC in children receiving vitamin D supplementation ($p < 0.05$) • After controlling confounding variables, only dt/ds associated with supplementation of vitamin D | • Lower caries incidence in those who received vitamin D supplementation • Children and mothers of higher education received vitamin D supplementation and had less caries • During periods of significant growth and development, children should take supplements |

**Table 3.** *Cont.*

| Authors Publication Year | Type of Study | Population | Age | Dentition | Results | Conclusions |
|---|---|---|---|---|---|---|
| Suárez-Calleja et al., 2021 [33] | Cohort study | 188 children | Dental examination 6–10 years of age | Mixed | • Levels of 25(OH)D below 20 ng/mL in mother and at 8 years of age is a risk factor for dental caries in children <br> • The risk of caries tripled when 25(OH)D were below 20 ng/ml | • An incorrect brushing technique and sugar consumption were the main reasons for caries in children <br> • Low levels of 25(OH)D during pregnancy have magnified the effect of cariogenic factors <br> • Supplementation of vitamin D for pregnant women and children could be an option |
| Chen et al., 2021 [38] | Cross-sectional | 1510 Chinese children | 44 +/− 8.2 months | Primary | • VDI and VDD leads to higher prevalence of ECC; 25.95%, 29.65%, respectively ($p = 0.016$) <br> • Negative correlation between 25(OH)D and number of caries (r = −0.103, $p < 0.001$) | • VDD, VDI increased the risk of ECC in preschool children |
| Beckett et al., 2022 [32] | Observational study | 81 children from New Zealand | Mean age of 6.6 years +/− 0.6 | Mixed | • Maternal vitamin D insufficiency (<50 nmol/L) during third trimester of pregnancy was associated with increased dental caries risk by 6 years of age (IRR = 3.55, $p < 0.05$) <br> • Vitamin D insufficiency was not related to enamel defect prevalence at any timepoint | • Maternal insufficiency of vitamin D during third semester of pregnancy is associated with higher caries experience in primary teeth by the age of 6 <br> • Recommendation for vitamin D supplementation during pregnancy and early life of the infant |

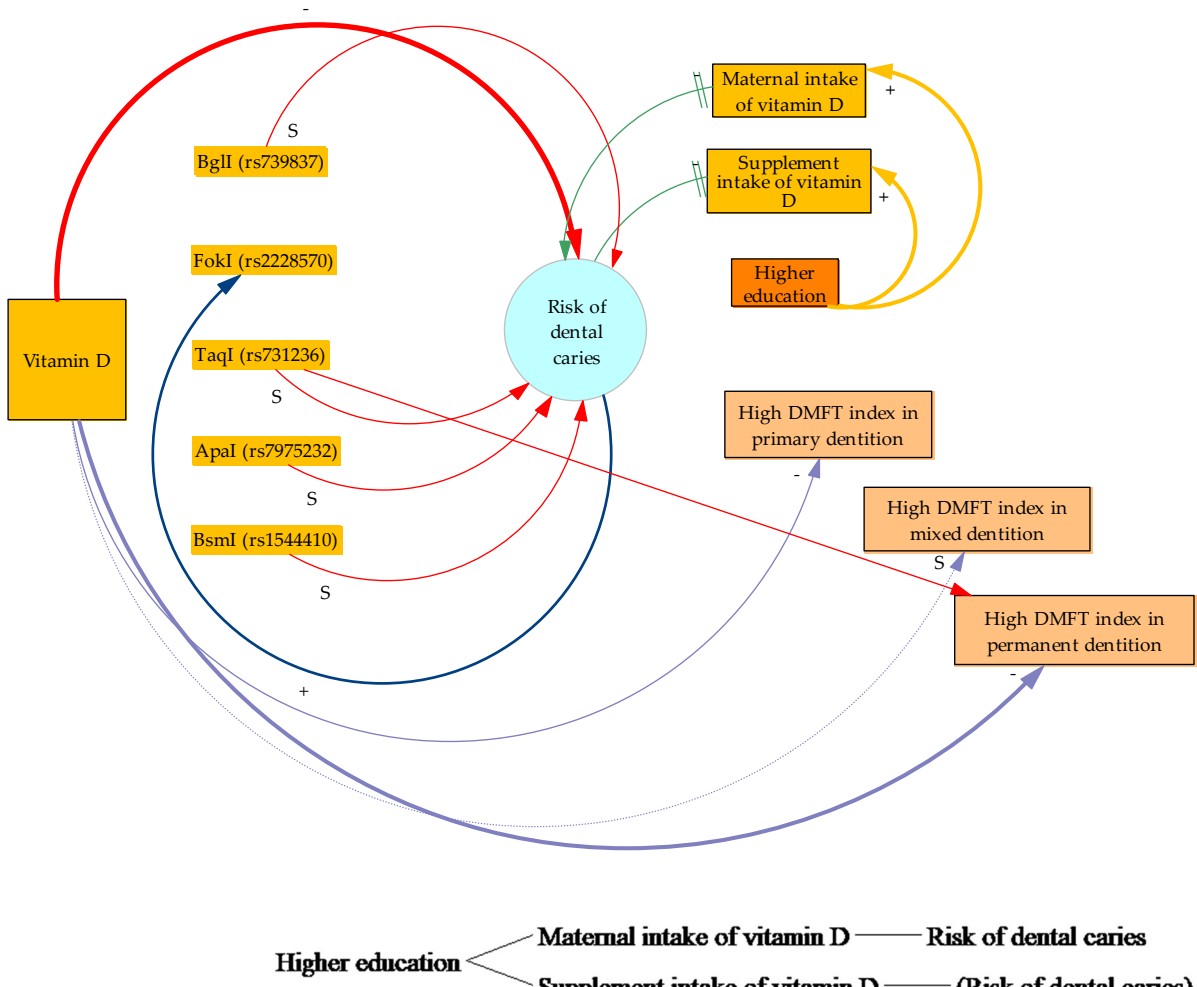

**Figure 2.** A Vensim diagram provides a graphical modeling interface with basic VDRPs and risk of dental caries. The diagram presents flows and causal loops among the basic factors affecting VD serum levels and risk of caries reported in our study. It also reveals a text-based system of equations in a declarative programming language. (+) positive or in tandem influence, (−) negative or inverse influence, (S) neutral relationship, (=) delayed response among factors, ( . . . ) border lines with dots support lack of sufficient data. Border lines support a quantitatively stronger relationship among factors. Red causal loops show the VDRP relationship with risk of dental caries; purple causal loops show VD serum levels and type of dentition; green ones show the way specific factors relate to risk of dental caries; orange ones show the interconnection between higher education and factors affecting dental caries; and blue is a reverse loop showing the interconnection of *FokI (rs2228579)* and risk of dental caries.

## 4. Discussion

Dental caries incidence can derive from a host of factors that may be related to the structure of dental enamel, immunologic response to cariogenic bacteria, or the composition of saliva. Dental caries causes a modern problem even in developed countries, leading to a decrease in the quality of life of the affected individuals and high economic costs for both individuals and society, with disparities related to well-known issues of socioeconomics, immigration, lack of preventive efforts, and dietary changes [58]. The burden of dental caries in children is incredibly high, causing pain that can affect school attendance, eating and speaking, and impair growth and development [21]. More data on the correlation of dental caries with VDD and polymorphisms were derived from studies conducted in children. However, it is not only in children that this problem could cause dysfunction in

social life and diet. Related problems can occur in young or middle-aged adults due to the cumulative and chronic nature of the disease [59]. Our study reveals the positive correlation of the DMFT index in permanent dentition with VDLs both in maternal and early birth stages. Our model also shows the importance of the *FokI (rs2228570)* polymorphism, in conjunction with the risk of dental caries, thus suggesting that relevant blood and genetic examinations can arise awareness of ECC, S-ECC, and high dental caries risk in permanent dentition, as already supported elsewhere [15].

As a higher risk of carious lesions has been associated with lower socioeconomic levels [60,61], we could expect further reductions in preventive care and frequency of dental visits [62]. According to the new Global Oral Health Status Report from the WHO [63], it is estimated that almost half of the world's population (45%, or 3.5 billion people) suffer from oral diseases such as caries, with 3 out of every 4 affected people living in low- and middle-income countries. Global cases of dental caries have increased by 1 billion over the last 30 years. This clearly indicates that many people do not have access to prevention measures and treatment of oral diseases such as dental caries. Socioeconomic factors should then be considered when planning preventive programs for caries. Higher education is a positive factor for controlling caries risk and an important part of the relevant model.

Thus, new ways of identifying high-risk individuals are future tools in controlling caries. As observed in our study, genetic variation in host factors such as VDPs may contribute to increased risks of the disease. Thus, it seems imperative that VDR gene polymorphisms can be proposed as a marker for identifying patients with high caries risk [21]. As mentioned by Di Spigna et al. [64], the genetic screening of VDRPs could be a valuable tool for the early identification of other health problems too, such as osteoporosis in female patients with rheumatoid arthritis (RA). Commercial kits, based on the Restriction Fragment Length Polymorphism (RFLP) method for VDR polymorphisms detection, could then be used in patients at high risk for dental caries too. It is proposed, then, that "the clinician and the lab manager may join to evaluate costs and availability, of the appropriate methods to setting molecular diagnostics of VDR Genotyping" [64]. Soon, genetic tests, now performed either by academic ultra-specialized labs or custom service laboratories using certified commercial kits [65], could be performed widely, in relation to demand in dental settings, under a protected insurance policy that could provide low-income citizens with cost-effective and accurate diagnostic and preventive tools for dental caries. In these cases, future studies should focus on the importance of the application of fluorides [66] or other remineralizing biomimetic materials [67] for prevention purposes.

This review highlights the need for more detailed and extensive studies to establish the cost and effectiveness of genotyping, introducing it more in everyday dental clinical practice in terms of caries prevalence. The limitations of the present study include the risk of misinterpreting data from clinical studies that have different methodology due to inadequate explanations and the possible incomplete retrieval of identified relevant research. In the future, with new genetic markers being identified and validated, dentists will have new ways and means to tailor specific dental therapy to individual caries risk genetic profiles [65]. Overall, this would reduce the final functional treatment costs within national health systems while further enhancing the economic status of lower- and middle-class individuals. Consequently, pharmaceutical and biotechnology companies must join their future investments to develop accurate and low-cost genetic tests for routine dental diagnostics.

## 5. Conclusions

This study strengthens the relationship of dental caries with VD levels, particularly in primary dentition and less in mixed and permanent dentition. VD levels in children are directly related to the maternal intake and socioeconomic factors and education of the mother.

There is an inconsistency among the case–control studies and their results regarding the different VDR gene polymorphisms and their influence on ECC and S-ECC for primary and mixed dentition and dental caries risk in permanent dentition. This fact is attributed to

the statistical heterogeneity between studies, the small number of existing relative studies, and the small sample sizes. There is a need for more well-conducted studies that will investigate other factors possibly influencing the current suggested model as well as the association between VDR gene polymorphisms and the risk of dental caries. VDR gene polymorphisms could be a marker for identifying not only children but also adult patients with high caries risk.

**Author Contributions:** Conceptualization, M.A., E.P. and C.R.; methodology, M.P. and M.A.; validation, M.P. and M.A.; formal analysis, M.P. and M.A.; investigation, M.P.; writing—original draft preparation, M.P. and M.A.; writing—review and editing, M.P., M.A., E.P., C.R. and T.V.; visualization, M.P., M.A. and E.P.; supervision, M.A., C.R. and T.V.; project administration, M.A. and T.V. All authors have read and agreed to the published version of the manuscript.

**Funding:** This research received no external funding.

**Institutional Review Board Statement:** Not applicable.

**Informed Consent Statement:** Not applicable.

**Data Availability Statement:** Not applicable.

**Conflicts of Interest:** The authors declare no conflict of interest.

## Abbreviations

Vitamin D (VD), vitamin D deficiency (VDD), vitamin D receptor (VDR), vitamin D receptor polymorphism (VDRP), Restriction Fragment Length Polymorphism (RFLP), Early Childhood Caries (ECC), Severe Early Childhood Caries (S-ECC), Decayed, Missing, and Filled Teeth (DMFT) index.

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
