# Peer review of "Vitamin D and Vitamin D Receptor Polymorphisms Relationship to Risk Level of Dental Caries"

_applsci, doi:10.3390/app13106014_

Round 1

Reviewer 1 Report

 Peponis et al has summarized the role of VitD in tooth formation and effects of VDR polymorphisms from published articles in scientific literature. However, there are similar and elaborate articles that already exists summarizing the role of vitD in tooth formation or effect of VDD and VDR gene polymorphisms on dental health (https://doi.org/10.3390/children8040302 ; https://doi.org/10.3390/nu12051471 ). The authors do not provide any new information or perform any analysis such as haplotyping or regression analysis in this review article to conclude. The result and discussion portion provides similar information, discussing the findings of existing literature.

For interest to the readers the authors should include further meta-analysis and graphical images for such a review article.

line 148-156: Why do the authors needs to elaborately define gene polymorphism?

Author Response

  • Dear reviewer, Figure 1 was added according to your comments. It is designed in Vensim software representing a model for the relationship between VD and VDRP with caries risk and factors affecting it, something that is newly introduced in the literature. You can see in lines 308-314. All changes in the manuscript are yellow underlined.

line 148-156: Why do the authors needs to elaborately define gene polymorphism?

We think it is important for the readers to bear in mind, it does not enlarge the extension of the text and helps the reader to understand better.

Dear reviewer

Thank you for your time and effort.

The authors

Reviewer 2 Report

Very interesting paper, very well organized, complete and easy to read.

It needs a mother tongue second reading.

Author Response

Dear reviewer, thank you for your comments. All changes are underlined in yellow. A model was further introduced as seen in added figure 1. A second reading was completed.

Reviewer 3 Report

A narrative review has been carried out on the effect of Vitamin D and Vitamin D Receptor Polymorphisms 2 on the prevalence of dental caries.

 Please add in brief regarding ECC and S-ECC in the introduction.

Fig 1- What do * and ** represent? Please add the reasons for excluding the articles

Author Response

-added. Please see lines 49-53

Fig 1- What do * and ** represent? Please add the reasons for excluding the articles

- we added: * Eligibility criteria were articles using: dmft-WHO diagnostic criteria with calibrated examiners, probability sampling, and sample sizes.** We excluded studies conducted on institutionalized patients.

Dear reviewer

Thank you for your time and effort.

The authors

Reviewer 4 Report

Comment to the authors:

Thank you for inviting me to review the study entitled “Effect of Vitamin D and Vitamin D Receptor Polymorphisms on Prevalence of Dental Caries: A Narrative Review”. This is supposed to be a narrative review looking at an association between vitamin D level and/or vitamin D receptor and risk of dental caries. 
I identified the following major flaws that need to be addressed before considering any possible publication. 
1) aim of the study was also to investigate an association between Vit D and tooth formation. However, I do not believe that such aim can be assessed with the current search terms used. I invite the authors to rethink whether including this aim in their study, or only discussing it on a general basis in the discussion section. 
2) the search has been conducted in a systematic fashion. However, the article is a narrative review. This means that in their results section, only a minority of articles are then discussed, which may lead to partial, incomplete or misleading information to the reader. I suggest the authors to turn this study into a systematic review, if they are able to justify the search based on comment 3 (below). Therefore, they should be able to summarize the available literature presented in the tables in aggregated way (such as: 3 studies claimed an association; however, 4 did not find any…. etc). 
3) the authors quote a meta-analysis published in 2021 conducted on the same topic. What does the current review add to that meta-analysis? If this topic has already been studies, why do we need another review that has lower quality (being a narrative review vs meta-analysis)? This undermines the need of the study itself. The authors need to find a strong justification to repeat the same search within only 2 years. 
4) in their presentation of the results the authors introduce the concept of the ECC. However, this has not been mentioned in the methods at all. Authors need to previously state that they will also search for effect of VIT D on ECC. It cannot be introduced in the results with no explanation in introduction nor methods 

The following comments identify specific points that need amendments. 

Abstract:
- minor mistakes are present (line 25)
- line 16: what does it mean “including several substances”? The ending of the sentence does not make much sense. A suggest omitting, and paraphs starting the following sentence with “among the possible substances implicated, vitamin D…”

Introduction
- same comment as above in line 36
- line 34: need citation 
- line 50:  instead of “to find”, I would replace with “to investigate / look for an association” 

Methods
- It is not clear why the authors limited the search only to records published after 2015
- the authors need to explicitly state the exclusion and inclusion criteria that justify the selection of the articles 
- does the search term includes the vitamin D receptor? 

Figure 1:
- there are two asterisks (* and **) that are not explained in the figure legend.  

Results:
- page 3. The first subsection of the results is “Vitamin D and tooth formation”. As explained above, I don’t think the current search term allows for a review of this topic. Moreover, the authors start discussion on a general basis what VDD cause on the skeleton. This is certainly not pertinent to the stated aim of the study. 
- the statement that vitamin D deficiency leads to rachitic tooth is only supported by reference 7. I suggest to include other studies supporting the same, otherwise this may be misleading for the reader and weakly supported. 
- the second big statement (line 137) is instead not supported by any reference. Moreover, “interglobular dentin” is not what is defined previously in the sentence (line 138). I suggest rephrasing the sentence. 
- line 165-167: the authors should speculate more on why some studies claim opposing results. This is the role of authors trying to critically understand and summarizing the available literature. Were the methods different? The population? The sample size did not permit for powered statement? 
- line 304: the authors comment on a systematic review published in 2013. however, the search only included papers from 2015 on. 
- pag 17: this section is confusing. The authors restate the results of the table; then they talk about ECC, then they go back discussion the results. The paragraphs of the results should be a summary of the evidence provided in the tables, not a mere restatement of what is already presented. 

Table 2:  
The table 2 might benefit from introducing a column / categorization of the results showing which type of dentition is analyzed (mixed vs deciduous vs permanent), as it seems that the results are contrasting depending on the type of dentition taken in consideration
- for a better understanding of what references are cited, it would be useful to name the references with the number of the list (e.g. Holla, put [23]). This is because it is not clear in the paragraph of page 4 (Vitamin D receptor gene polymorphisms) which references the authors are quoting among those cited in the table. For example, in line 162, they state that “there is a strong correlation between VDR gene SMPs and mineral density”, and they quote 16 and 17. This means that all the other references quoted in Table 2 does not support? If that is the case, the statement is supported by the minority of the studies, which may change the conclusion. 

Discussion

- I suggest omitting lines 323-324 on COVID-19. COVID-19 has really nothing to do with this topic. 

- the discussion should restate the aim of the study and briefly summarize the results found in 1-2 sentences. This is lacking in the current version of the manuscript 

Conclusions:

- The conclusions are not a logic consequence of the results. First of all, the conclusions do not mention at all the relationship of dental caries with VIT D level, but only mention that there is inconsistency with VDR gene polymorphisms. Second of all, nothing leads the reader to think that this is the conclusion supported by the results, as the results lack a aggregated summary of the evidence. 

Author Response

Authors response

-Aim of the study was rewritten. All changes are yellow underlined. We emitted association between VD and tooth formation from our scope

2) the search has been conducted in a systematic fashion. However, the article is a narrative review. This means that in their results section, only a minority of articles are then discussed, which may lead to partial, incomplete or misleading information to the reader. I suggest the authors to turn this study into a systematic review, if they are able to justify the search based on comment 3 (below). Therefore, they should be able to summarize the available literature presented in the tables in aggregated way (such as: 3 studies claimed an association; however, 4 did not find any…. etc).

-We rewritten the specific paragraphs according to suggestions. See yellow underlined text.

3) the authors quote a meta-analysis published in 2021 conducted on the same topic. What does the current review add to that meta-analysis? If this topic has already been studies, why do we need another review that has lower quality (being a narrative review vs meta-analysis)? This undermines the need of the study itself. The authors need to find a strong justification to repeat the same search within only 2 years.

-We added in the introduction

As awareness of VDD has increased among patients and the health community, many authors have conducted clinical trials to find an association between VDD and dental caries.[12-14] In addition, there is a systematic review investigating vitamin D receptor (VDR) gene polymorphisms in relation to increased risk of dental caries in children.[15] The aim of this systematic review was to further investigate the relation between VDD, VDRP, ECC and S-ECC in children (primary and mixed dentition) and dental caries risk in adults (permanent dentition). Another scope was to design a model incorporating factors and interactions that address this relationship.

4) in their presentation of the results the authors introduce the concept of the ECC. However, this has not been mentioned in the methods at all. Authors need to previously state that they will also search for effect of VIT D on ECC. It cannot be introduced in the results with no explanation in introduction nor methods

-As mentioned before we included this suggestion in our scope below

As awareness of VDD has increased among patients and the health community, many authors have conducted clinical trials to find an association between VDD and dental caries.[12-14] In addition, there is a systematic review investigating vitamin D receptor (VDR) gene polymorphisms in relation to increased risk of dental caries in children.[15] The aim of this systematic review was to further investigate the relation between VDD, VDRP, ECC and S-ECC in children (primary and mixed dentition) and dental caries risk in adults (permanent dentition). Another scope was to design a model incorporating factors and interactions that address this relationship.

The following comments identify specific points that need amendments.

Abstract:

- minor mistakes are present (line 25)

-we changed the whole abstract

- line 16: what does it mean “including several substances”? The ending of the sentence does not make much sense. A suggest omitting, and paraphs starting the following sentence with “among the possible substances implicated, vitamin D…”

-we changed the whole abstract

Introduction:

- same comment as above in line 36 -Corrected

- line 34: need citation -Corrected

- line 50:  instead of “to find”, I would replace with “to investigate / look for an association”

-Corrected

Methods:

- the authors need to explicitly state the exclusion and inclusion criteria that justify the selection of the articles

-Added both in the abstract and in the materials and methods sections

- does the search term includes the vitamin D receptor?

-corrected

Figure 1:

- there are two asterisks (* and **) that are not explained in the figure legend. 

-Explanation was added both in the text and in the figure

Results:

- page 3. The first subsection of the results is “Vitamin D and tooth formation”. As explained above, I don’t think the current search term allows for a review of this topic. Moreover, the authors start discussion on a general basis what VDD cause on the skeleton. This is certainly not pertinent to the stated aim of the study.

-corrected. Please see yellow underlined text in the relevant paragraph

- the statement that vitamin D deficiency leads to rachitic tooth is only supported by reference 7. I suggest to include other studies supporting the same, otherwise this may be misleading for the reader and weakly supported.

-added

- the second big statement (line 137) is instead not supported by any reference. Moreover, “interglobular dentin” is not what is defined previously in the sentence (line 138). I suggest rephrasing the sentence.

-corrected

- line 165-167: the authors should speculate more on why some studies claim opposing results. This is the role of authors trying to critically understand and summarizing the available literature. Were the methods different? The population? The sample size did not permit for powered statement?

-corrected

- pag 17: this section is confusing. The authors restate the results of the table; then they talk about ECC, then they go back discussion the results. The paragraphs of the results should be a summary of the evidence provided in the tables, not a mere restatement of what is already presented.

-corrected

Table 2: 

The table 2 might benefit from introducing a column / categorization of the results showing which type of dentition is analyzed (mixed vs deciduous vs permanent), as it seems that the results are contrasting depending on the type of dentition taken in consideration

-done, underlined with yellow in Table 2

- for a better understanding of what references are cited, it would be useful to name the references with the number of the list (e.g. Holla, put [23]). This is because it is not clear in the paragraph of page 4 (Vitamin D receptor gene polymorphisms) which references the authors are quoting among those cited in the table. For example, in line 162, they state that “there is a strong correlation between VDR gene SMPs and mineral density”, and they quote 16 and 17. This means that all the other references quoted in Table 2 does not support? If that is the case, the statement is supported by the minority of the studies, which may change the conclusion.

-corrected

Discussion:

- I suggest omitting lines 323-324 on COVID-19. COVID-19 has really nothing to do with this topic.

-corrected

- the discussion should restate the aim of the study and briefly summarize the results found in 1-2 sentences. This is lacking in the current version of the manuscript

-corrected

Conclusions:

- The conclusions are not a logic consequence of the results. First of all, the conclusions do not mention at all the relationship of dental caries with VIT D level, but only mention that there is inconsistency with VDR gene polymorphisms. Second of all, nothing leads the reader to think that this is the conclusion supported by the results, as the results lack a aggregated summary of the evidence.

-corrected. We added:

This study strengthens the relationship of dental caries with VD level, more in the primary and less in the mixed and permanent dentition. VD level in children is directly related to the maternal intake and socioeconomic factors and education of the mother.

There is an inconsistency among the case-control studies and their results about the different VDR gene polymorphisms and their influence on ECC and S-ECC for primary and mixed dentition and dental caries risk in permanent dentition. This fact is attributed to the statistical heterogeneity between studies, the small number of existing relative studies, and the small sample sizes. There is a need for more well-conducted studies that will investigate more factors possible to influence the current suggested model as well as the association between VDR gene polymorphisms and the risk of dental caries. VDR gene polymorphisms could be a marker for identifying not only children but also adult patients with high caries risk.

Dear reviewer

Thank you for your time and effort to review our manuscript.

We addressed all issues suggested the best way possible.

The authors

Round 2

Reviewer 1 Report

Though the context of the article has improved, but there is no improvement on the analytical content of the article. The authors have included a 'Vensim' model, unfortunately I am not an expert to review this figure. However, it is very confusing as there is no clear description of the figure in the text, such as, what is the message readers get from this diagram, what was the input data for obtaining this model, what do the color coding of the arrows means etc. 

Author Response

Following the reviewer’s suggestions, we presented the vensim model in a detailed way. Further modifications in the article are highlighted in green color. Vensim diagram is now clearly explained in figure 2.

We hope we met expectations

Thank you for your time and effort.

The authors

Reviewer 4 Report

The authors have provided an extensive review. Unfortunately, it is not enough to indicate that a systematic review was conducted. A systematic review requires a PICO question, the quality assessment, the collection of the results, identification of which data will be collected, etc. The current study totally lacks the methodology necessary to call the search a systematic review. Here the authors only changed the name from "literature" to "systematic" review, without applying and performing everything that is expected with the systematic review. 

I also still don't agree with the fact that in the results section, instead of limiting the section to presenting the results of the included studies, the authors provide lengthy explanation of the topic (e.g., biological activity of VD, explanation of polymorphisms, ...) 

Minor correction: line 80, pag 2. When the authors indicate the presence of the systematic review, they have to highlight what it is still lacking in the literature, that justifies the current research. Especially if no clinical trials have been conducted from the publication of that SR to now.  

IN conclusion, at the current state the manuscript cannot be published, because it claims to be a SR, without having the methodology of it. Either the authors go back to the original plan of literature review, or they have to include the necessary methodology of SR.  

Author Response

Authors’ response= Further alterations in the article are highlighted in green color. The “systematic review” title was suggested by another reviewer, and we changed it accordingly following your suggestions. We do agree that this is a narrative review that plans to highlight some factors influencing the relationship between VD, VDP and dental caries. We have made accordingly the necessary change.

I also still don't agree with the fact that in the results section, instead of limiting the section to presenting the results of the included studies, the authors provide lengthy explanation of the topic (e.g., biological activity of VD, explanation of polymorphisms, ...)

Authors’ response= We have further eliminated the text and kept only necessary information

Minor correction: line 80, pag 2. When the authors indicate the presence of systematic review, they have to highlight what it is still lacking in the literature, that justifies the current research. Especially if no clinical trials have been conducted from the publication of that SR to now. 

Authors’ response= we have highlighted accordingly. Further in the text we describe better the Vensim diagram, in a more analytical way.

IN conclusion, at the current state the manuscript cannot be published, because it claims to be a SR, without having the methodology of it. Either the authors go back to the original plan of literature review, or they have to include the necessary methodology of SR. 

Authors’response= we have corrected accordingly (see comment before)

We hope we met expectations

Thank you for your time and effort.

The authors

Round 3

Reviewer 4 Report

The authors have provided a third revision of their article. I believe that the current version of the manuscript is now acceptable for publication. It does not convey any misleading information, it is decently structured, it does not imply the methodology of a systematic review (where many parts were missing). I am satisfied with the current product provided by the authors.   

Author Response

Dear reviewer

Thank you for your time and effort to help making better our manuscript

The authors
